# Effects of Casein Hydrolysate Prepared with Savinase on the Quality of Bread Made by Frozen Dough

**DOI:** 10.3390/foods12203845

**Published:** 2023-10-20

**Authors:** Hatice Bekiroglu, Gorkem Ozulku, Osman Sagdic

**Affiliations:** 1Department of Food Engineering, Faculty of Chemical and Metallurgical Engineering, Yildiz Technical University, Istanbul 34220, Turkey; h.bkroglu@gmail.com (H.B.); osagdic@yildiz.edu.tr (O.S.); 2Food Engineering Department, Agricultural Faculty, Şırnak University, Şırnak 73300, Turkey

**Keywords:** casein hydrolysate, frozen dough, bread quality

## Abstract

The effect of casein savinase hydrolysate (CSH) usage on frozen dough (1%, 1.5% and 2%, g/100 g flour) was investigated in terms of rheological, thermal and structural characteristics of wheat doughs and the textural and color properties of corresponding breads. Rheological measurements showed that CSH addition into dough led to a reduction in G′ and G″ values, but a similar trend was not observed in frozen dough samples. The increase in protein band intensity was observed for control dough (CD) after frozen storage (−30 °C, 28 days), while there were no increases in the band intensities of the doughs with CSH. The freezable water content of unfrozen doughs decreased gradually with the addition of CSH, dependent on concentration level. Frozen storage caused a notable reduction in the α-helices structure of the CD sample (*p* < 0.05) while no significant variation was observed for the doughs containing CSH (*p* > 0.05). The lowest specific volume reduction and hardness increment were observed for the breads containing 1.5% and 2% CSH. Frozen storage caused a significant reduction in the b* value of bread crust (*p* < 0.05), while no significant effect was observed for L* and a* value during frozen storage (*p* > 0.05). Overall, CSH incorporation into frozen dough can be an alternative that could reduce the quality deterioration of frozen bread.

## 1. Introduction

Milk proteins are frequently preferred in the development of biological characteristics of foods because of their high digestibility features and abundant essential amino acid content [1]. They are also considered to be a fundamental dietary component due to their balanced and adequate nutrient content. In addition to the superior bioactive properties they provide, milk proteins also enable technological characteristics such as emulsion, oil–water retention capacity and texture [2,3].

In recent years, the importance of healthy and functional ingredients has become more widely accepted, leading to an increased demand for improving the functional characteristics of frequently consumed nutrients. This demand has generated food products with peptides containing much more biologically qualified and active components. Enriching food items with low protein content, such as bakery products, and especially bread, with milk protein and its hydrolysates can significantly increase their nutritional value [4]. The enzymatic hydrolysis-based modification of milk proteins has significant potential as a tool in food protein processing to optimize the technofunctional, biological and nutritional properties of proteins [5]. Casein hydrolyzed formulations have been sold for many years due to their exceptional nutritional content, amino acid composition, commercial availability in huge numbers, and affordable price [6]. Casein hydrolysates can be used to enhance the nutritional value, stability, shelf life, and many other characteristics of various food products, as well as protein supplements [7].

The use of frozen dough in bread and bakery products has been increasing. Basic factors like shelf life and bakery quality may be regulated with the use of frozen dough, and at the same time it enables the production of huge quantities of dough, ease of storage and transportation, and the production of fresh products in small and medium-sized enterprises [8]. It is known that the freezing and thawing process in the production of frozen bread damages the gluten network [9] and prevents the bread from rising, thus causing low volume and harder crumb formation [10,11]. Some proteins and their hydrolysates have been used in several studies to eliminate these quality deteriorations. The effects of milk protein [12], pigskin gelatin [13] and sweet potato protein hydrolysates [14] on the quality of breads made from frozen dough have been investigated. In a study of Zhou et al. [15], desalted egg white and gelatin mixture systems provided a positive effect on the quality of frozen dough by slowing down the increase in the freezable water content and the loss tangent (tanδ) value. Also, a cryo-protective effect of ice-binding peptides derived from collagen hydrolysates was reported by Cao et al. [16].

Additionally, casein hydrolysate could be an alternative for frozen dough quality improvements thanks to its inexpensive and efficient production. Therefore, we aimed to investigate the contributions of casein hydrolysates produced from savinase to frozen dough and corresponding bread quality in this study. To realize this aim, the rheological, thermal and protein analyses (SDS-PAGE and secondary structure) were studied alongside the quality of breads made from frozen dough.

## 2. Materials and Methods

### 2.1. Materials

Wheat flour (13.5% moisture, 0.73% ash, 10.8% protein and 58% water absorption) was purchased from Eksun Food Industry Co., Ltd. (Tekirdağ, Turkey). Dry yeast (Pakmaya) and salt (Billur) were purchased from a local market. The bovine casein (NaCas) was obtained from a local brand (Milkon, Çallı Food Food Industry and Trade Inc., İstanbul, Turkey). The savinase enzyme (protease from *Bacillus* sp., 16 U/g), sodium dodecyl sulfate (SDS), acrylamide, and all other chemicals and solvents were purchased from Sigma-Aldrich (St. Louis, MO, USA).

### 2.2. Preparation of Casein Hydrolysate

Casein was first dissolved with distilled water and kept at +4 °C for 10 h for complete hydration. The pH of the casein solutions was adjusted to pH 9, which is optimum for the savinase enzyme using 0.1 N NaOH. Then, the volume of casein solution was completed with distilled water to 5 g protein/100 mL. Savinase was added to the casein solutions (enzyme:substrate, 1:100) and hydrolyzed for 180 min in a water bath set at 50 °C. During the hydrolysis period, the pH of the solution was adjusted to 9 by using 0.1 N NaOH every 15 min, and the hydrolysis degree (HD) of casein was determined according to the total amount of base consumed and the principle of the pH-stat method [17,18,19]. The hydrolysis degree of casein hydrolysates was obtained as 24.7%. At the end of the hydrolysis period, the enzyme inactivation of the casein solutions was achieved by keeping them in boiling water for 15 min. They were centrifuged at 6500 rpm for 15 min and the supernatant fraction was freeze-dried.

### 2.3. Preparation of Bread Dough and Baking

The bread dough formulation (control) consisted of 100 g wheat flour, 2 g dry yeast, 1.5 g salt and 58 g water. The wheat flour was replaced by casein savinase hydrolysate (CSH) at the concentrations of 1%, 1.5% and 2% wheat flour dry basis (*w*/*w*) to produce hydrolysate-added dough (HD). All ingredients were kneaded in a dough mixer (KitchenAid, Benton Harbor, MI, USA) at speed 4 for 5 min. After complete mixing, the dough was sheeted, molded and panned. The unfrozen dough samples were fermented at 30 °C and 85% relative humidity (RH) for 2 h and baked at 235 °C for 25 min in an electrical oven (Maksan, Nevşehir, Turkey). The frozen dough samples were directly stored at −35 °C for 28 days after shaping. The thawing process was performed at 4 °C for 3 h. After fermentation (30 °C and 85% RH for 2 h) of the thawed dough, the same baking procedure was applied.

### 2.4. Rheological Measurements

Unfrozen and thawed dough samples were subjected to dynamic rheological tests by using stress and temperature controlled rotational rheometer (Antonpaar MCR 302, Graz, Austria). Frequency sweep tests were conducted between 0.1 rad/s and 100 rad/s with a fixed strain of 0.1% in the linear viscoelastic region at 25 °C with a 2 mm gap. Two replications of each sample were carried out.

### 2.5. Freezable Water (FW) Content

The FW content of dough samples was determined with the differential scanning calorimeter (DSC; TA instrument Q20, USA) by using the method of Chen et al. [14]. Briefly, 10 mg of the dough sample was sealed hermetically in an aluminum tray and an empty aluminum tray was used as the reference. The following heating program was applied: cooling from 20 °C to −40 °C at a rate of 5 °C/min and keeping constant for 5 min, and then heating from −40 °C to 10 °C at a rate of 5 °C/min. The enthalpy (ΔH) of the melting peak was determined during the heating process and FW content (%) was calculated according to Equation (1):(1)$$FW\left(\%\right)=\frac{\Delta H}{\Delta Ho\times Wc}\times 100$$
where ΔH is the enthalpy (J/g) of the melting peak of the endothermic curve; ΔHo is the enthalpy (334 J/g) of melting peak of pure water; and Wc represents the total water content (%) of the dough, which was measured using a moisture analyzer (Radwag MR50, Poznań, Poland). Two replications of each sample were performed.

### 2.6. Sodium Dodecyl Sulfate–Polyacrylamide Gel Electrophoresis (SDS-PAGE) Analysis

SDS-PAGE analyses of all dough samples were carried out according to the method described by Laemmli [20]. First, the freeze-dried dough samples were powdered via a grinder. Powder dough samples were mixed with sample buffer at 10 mg/mL and kept in a boiling water bath for 5 min. The samples were added to each loading well as 20 µL and sorted according to their molecular weights by running under a 20 mA/gel electric current in the separation gel (7.5% and 20% acrylamide) using a vertical electrophoresis system (Mini-PROTEAN^®^System, Bio-Rad, Hercules, CA, USA). After the electrophoresis process, the bands stained with staining solution (comassie brillant blue:methanol:acetic acid:water; 0.25%:50%:10%:39.75%) were removed from excess dye by keeping them in the destaining solution (methanol:acetic acid:water; 50%:10%:40%) and then visualized with the Biorad Gel Doc EZ imaging system.

### 2.7. Fourier Transform Infrared Spectroscopy (FT-IR)

The secondary structure of dough samples was determined by using a Bruker Tensor 27 spectrometer equipped with a DLa TGS detector (Bremen, Germany). FT-IR spectra of all samples were obtained with wave numbers from 400 to 4000 cm^−1^ during 16 scans per spectra, with 2 cm^−1^ resolution. The measurements were carried out 3 times. Deconvolution was applied to all spectra by the interpretation of changes in the overlapping amide I band (1600–1700 cm^−1^) using Origin 2020b software [21].

### 2.8. Determination of Bread Quality

#### 2.8.1. Specific Volume (SV)

The volume (mL) of bread was measured in duplicate by using the rapeseed displacement method (Method 10-05, AACC 2000). The SV of breads was calculated by dividing the volume by the weight (g) of bread.

#### 2.8.2. Texture

Each bread sample was cut into slices that were 1.25 mm thick after cooling for 2 h at room temperature. Texture Profile Analysis (TPA) was performed by using a texture analyzer (SMS TA.XT2 Plus, Glasgow, UK) equipped with a 5 kg load cell and 36 mm diameter cylindrical compression probe. The TPA test was conducted with 50% compression, 5.0 mm/s test speed and 5 s delay time between the two compression cycles. Duplicate measurements were performed.

#### 2.8.3. Color

The crust and crumb color of bread samples were measured by using a Chromameter (CR-100 Konica Minolta, Tokyo, Japan), recording the lightness (L*), redness (a*) and yellowness (b*) values. All determinations were carried out in triplicate. Three measurements were taken from each sample. The color differences (∆E) were determined by using the following Equation (2):(2)$$∆E=\sqrt{\left(L_{0}^{*}-L_{}^{*}\right)+\left(a_{}^{*}-a_{0}^{*}\right)+\left(b^{*}-b_{0}^{*}\right)}$$
where L_0_^*^, a_0_^*^ and b_0_^*^ belong to control bread, which is a reference sample, while L*, a* and b* belong to bread containing CSH.

### 2.9. Statistical Analysis

An analysis of variance (ANOVA) followed by Duncan’s multiple range tests was used to determine any significant differences (*p* < 0.05) between the mean values. SPSS version 19.0 software (SSPS Inc., Chicago, IL, USA) was used.

## 3. Results and Discussion

### 3.1. Dynamic Rheological Properties of Dough

Figure 1 exhibits the storage (G′) and loss (G″) modulus of dough samples, indicating the elastic and viscous properties, respectively. The loss factor, tan δ, was also shown to reflect the viscoelasticity of samples. For all samples, the G′ was greater than G″, showing that the dough was more elastic and exhibited solid-like behavior [22]. CSH addition into the dough led to a reduction in G′ and G″ values, but a similar trend was not observed in frozen dough samples. Frozen storage decreased the G′ and G″ values of all dough samples, and control dough (CD) had the lowest value. The same behavior was shown in G″ values. Also, the decreasing effect of frozen storage on dough containing 2% was lower than that in other doughs with CSH (1% HD and 1.5% HD) (Figure 1B). These results indicated that CSH addition caused a detrimental effect due to lowering G′ and G″ and increasing tan δ values in a fresh dough system. According to the study of Khatkar et al. [23], the flour from wheat, showing good bread quality, had a higher G′ value and lower tan δ. On the other hand, the addition of CSH improved the gluten strength of frozen dough when compared to control-frozen dough (CFD), as a result of elevated G′ and G″ values [24]. A similar observation was reported by Zhou et al. [15] for a frozen dough system containing a desalted egg white and gelatin mixture. In the dynamic rheological measurement of the frozen dough system, tan δ value (G″/G′) is considered an important parameter to monitor the viscoelasticity of dough [22]. The tan δ values obtained in this study were lower than 1, indicating that the elasticity of the samples was greater than the viscosity [15]. Frozen storage resulted in a remarkable increase in tan δ for all samples (Figure 1C). The increment of the tan δ value during freezing has previously been reported by many studies [14,15]. This was attributed to the gradual increase in ice crystals during freezing, causing damage to the gluten network and subsequently resulting in an increased viscosity of the dough. The rise of tan δ in dough with CSH was lower than CD after frozen storage (Figure 1C). This can be explained by slowing the ice crystal growth due to the addition of CSH.

### 3.2. SDS-PAGE Analysis

The changes in SDS-PAGE patterns of proteins with the incorporation of casein hydrolyzate during frozen storage are presented in Figure 2. The formation of new bands and/or changes in band intensities showed polymerization/depolymerization behavior in the gluten network [25,26,27]. Depolymerization exists in the protein matrix of wheat dough when exposed to frozen conditions [26]. The increase in protein band intensity was observed for control dough (CD) after frozen storage (CFD), indicating that a depolymerization of the high molecular weight glutenin subunits (HMW-GS) region occurred. Similar results were previously reported by numerous studies for wheat dough under frozen conditions [13,28,29]. As for the samples including casein hydrolyzate, no depolymerization was observed due to frozen storage as there was no increases in band intensities, except for the 2% HD sample (Figure 2B). The 2% casein hydrolyzate addition increased the band intensity after frozen storage, showing that this level of addition may inhibit the crosslinking of SDS-insoluble proteins and interact with SDS-soluble proteins [13]. The incorporation of unhydrolyzed pigskin gelatin at 1% level also resulted in an increase in band intensity corresponding to the ω-gliadins region in a study by Yu et al. [13], who suggested that pigskin gelatin contributed to the formation of aggregates through non-covalent interactions. The inclusion of casein hydrolyzate in the wheat dough matrix led to a decrease in band intensities in the HMW-GS region when compared to CD. This can show that new interactions between glutenin, casein hydrolyzate and water were formed during dough processing since band intensity increment was observed in the low molecular weight glutenin subunit (LMW-GS), γ-gliadin and α/β gliadin regions. Another possible explanation of this can be the formation of protein polymers with a larger size, which were unable to migrate into the resolving gel [25]. Also, new band formation due to the addition of casein hydrolyzate into wheat dough was not observed in the present study, indicating the absence of new protein interactions. Similar results were reported in a study by Yu et al. [13], who incorporated the pigskin gelatin into frozen dough, and a study which investigated the effect of natural inulin on frozen dough [27].

### 3.3. Freezable Water Content

Freezable water (FW) in dough produces ice crystals when subjected to freezing. The crystallization amount of ice is directly attributed to the FW content in dough and causes the weakening of the gluten structure. Therefore, it is important to measure FW content to monitor the quality of dough during frozen storage [15,24]. As shown in Figure 3, the FW (%) content of dough samples increased after frozen storage (−30 °C, 28 days) due to the formation of ice crystals. The rheological properties of dough are also influenced by FW content since ice crystals disrupted the gluten network. This was supported by the tan δ value of dough samples after frozen storage (Figure 1C) due to an increasing trend being shown. CSH incorporation gradually decreased the FW (%) content of fresh dough. This could be due to the binding of water molecules to CSH, as stated in a study by Bekiroglu et al. [7]. Also, the increasing effect of frozen storage on FW (%) was suppressed by the addition of CSH, dependent on concentration (Figure 3). The 2% HD sample exhibited the lowest FW (%) value, suggesting that a 2% CSH substitution level could change the moisture distribution within the dough [29]. These results are in line with the study carried out by Chen et al. [14], who investigated the effect of sweet potato protein hydrolysates on the quality of frozen dough. The conversion of bound water to freezable water was also inhibited when a desalted egg white and gelatin mixture was added into frozen dough in a study by Zhou et al. [15].

### 3.4. Analysis of Secondary Structures of Protein

The secondary structures of gluten proteins corresponding to the amide I region (1700–1600 cm^−1^) are shown in Table 1. The FTIR spectra of the samples after baseline correction and Gaussian smoothing are shown in Figure 4. Absorption peaks were ascribed as β-sheets (1682–1696 cm^−1^), β-turns (1662–1681 cm^−1^), α-helices (1650–1660 cm^−1^) and random coils (1640–1650 cm^−1^) based on the reports of Yang et al. [30] and Zhou et al. [15]. It was reported that α-helices and β-sheets were relatively orderly and stable, while β-turns and random coils were disordered [24]. The incorporation of CSH into wheat dough formulation significantly affected the secondary structure of proteins, except for the β-sheets structure at Day 0. Frozen storage (−30 °C, 28 days) caused a notable reduction in the α-helices structure of the CD sample (*p* < 0.05), while no significant variation was observed for the samples containing CSH (*p* > 0.05). This could be due to the inhibition of recrystallization by CSH addition in the frozen storage state. The structure of β-turns significantly increased after frozen storage, indicating a transformation of α-helices into β-turns structure during frozen storage [15]. The reduction in α-helices and increment of β-turns were higher in CD than the doughs with CSH. The random coils structure was not changed during frozen storage (*p* > 0.05), similar to the study of Yu et al. [13], who investigated the effect of pigskin gelatin in frozen dough.

### 3.5. Specific Volume, Hardness and Color of Bread

The quality of bread is mostly evaluated in terms of specific volume (SV), hardness (N) and color characteristics. These criteria are also important indicators that are used for monitoring the frozen dough quality. Figure 5 presents the SV (mL/g) and hardness (N) of bread samples. CSH addition into bread formulation had no remarkable effect on bread quality (*p* > 0.05). The primary impact of frozen storage at −30 °C for 28 days on the bread was a reduction in SV values and an increase in hardness. These phenomena, which have previously been reported in many studies, suggest that during frozen storage ice crystals lead to a decrease in the gas-holding capacity of the dough by destroying the structure of the gluten [14,15]. Also, Cao et al. [16] noted that the fermentation capacity of the dough was influenced by the growth of ice crystals, which decreased the survival of yeast cells. However, the significant effect was shown only in CD and 1%HD after frozen storage (*p* < 0.05). The lowest SV reduction and hardness increment were observed for the breads containing 1.5% and 2% CSH. This can be attributed to the FW (%) content of 1.5%HD and 2%HD samples also being lower than in the CD sample (Figure 3). Mao et al. [31] reported that a reduction in the FW content in frozen dough protected the gluten network by preventing the formation of ice crystals.

The effect of CSH addition and frozen storage (−30 °C, 28 days) on the crust and crumb color of bread samples is shown in Table 2. CSH addition decreased the L* value of the bread crust and a significant reduction was observed for the breads containing 1.5% and 2% CSH when compared to control (*p* < 0.05). The browning of the bread crust (decrease in L* value) with the addition of CSH as a protein source showed an intensive Maillard reaction compared to the control. In a study which investigated the effect of pigskin gelatin in frozen bread, it was indicated that the browning of the bread crust was due to the addition of pigskin gelatin [13]. Frozen storage caused a significant reduction in the b* value of bread crust (*p* < 0.05), while no significant effect was observed for L* and a* value during frozen storage (*p* > 0.05). Similar results for the b* value of the bread crust were also reported by Sharadanant and Khan [32]. In terms of color differences (ΔE), frozen storage had no significant effect on the ΔE value of the crust color, but the ΔE values of 1.5% HFD and 2% HFD were higher than those of other samples (e.g., CFD and 1% HFD). As for the crumb color, frozen storage resulted in the reduction in L* value in the crumb (*p* < 0.05). A similar trend was also noted by Yu et al. [13] at the end of 60 days of frozen storage (−18 °C). On the other hand, Shon et al. [12] reported that no significant effect was observed for crumb L* value during frozen storage (−20 °C, 60 days). The incorporation of CSH into fresh bread increased the b* value of the bread crumb (*p* < 0.05). While frozen storage increased the crumb b* value of the control fresh bread (*p* < 0.05), the breads with CSH were not affected by frozen storage (*p* > 0.05) in terms of crumb b* value. The bread made of control-frozen dough (CFD) exhibited the highest color differences (ΔE) in bread crumb when compared to the frozen dough breads containing CSH. This shows that frozen storage caused no significant effect on the breads with CSH in terms of ΔE in the bread crumb (*p* < 0.05).

## 4. Conclusions

In this study, casein hydrolysate prepared with savinase (CSH) was used to improve the quality of frozen dough. The frozen dough with CSH did not demonstrate any notable alteration in the structure of α-helices following frozen storage at −30 °C for 28 days. The results obtained from the dynamic rheological measurement and freezable water content showed that the incorporation of CSH into frozen wheat dough slowed down the ice crystals’ growth. These findings resulted in an amelioration in the bread quality of 1.5% and 2% CSH-containing samples since the lowest specific volume reduction and hardness increment were exhibited. In summary, casein hydrolysate obtained from savinase can be considered as an alternative bakery ingredient for frozen dough systems.

## Figures and Tables

**Figure 1 foods-12-03845-f001:**
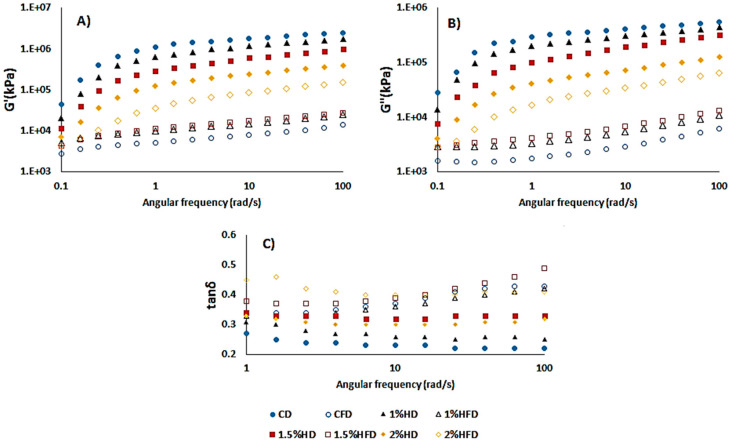
Effect of casein savinase hydrolysate (CSH) addition on dynamic rheological properties of dough samples. (**A**) Storage modulus, G′, (**B**) Loss modulus, G″, (**C**) Loss factor, tanδ. CD: control dough, CFD: control-frozen dough, HD: dough containing CSH, HFD: frozen dough containing CSH.

**Figure 2 foods-12-03845-f002:**
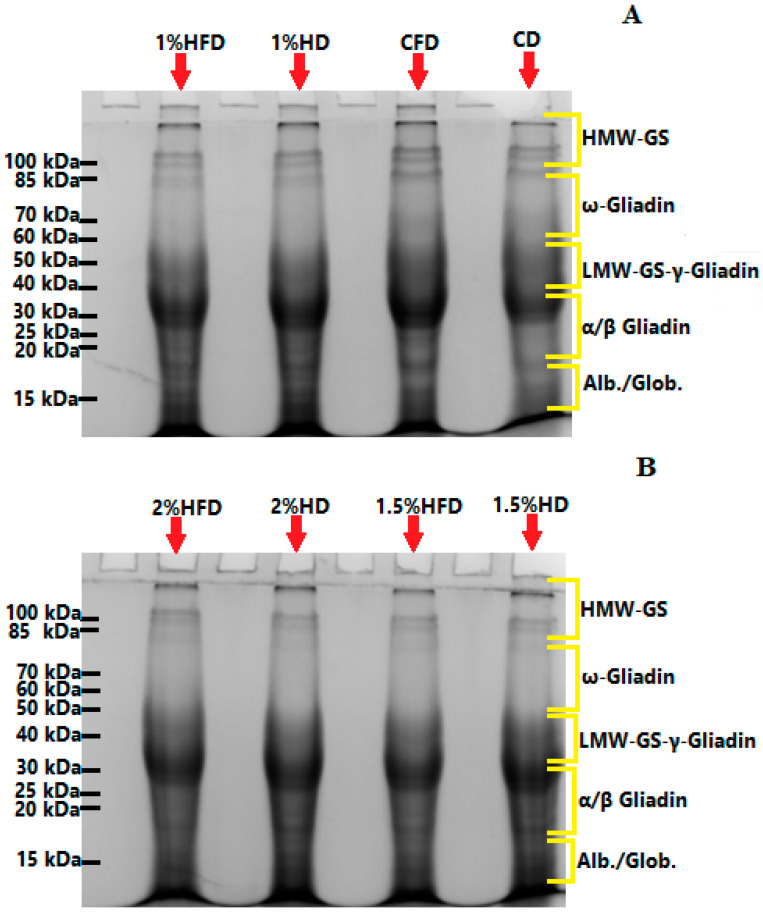
(**A**) SDS-PAGE patterns of control dough and the doughs containing casein savinase hydrolysate (CSH) at the level 1%. (**B**) SDS-PAGE patterns of the doughs containing CSH at the level 1.5% and 2% CD: control dough, CFD: control-frozen dough, HD: dough containing CSH, HFD: frozen dough containing CSH. The molecular weights of the dough samples were arranged in the bands.

**Figure 3 foods-12-03845-f003:**
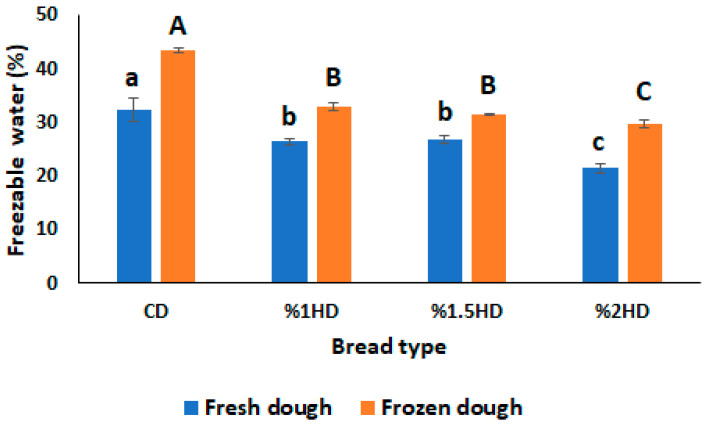
Effect of casein savinase hydrolysate (CSH) addition on freezable water content in dough samples. CD: control dough, HD: dough containing CSH. Different lowercase letters show statistically significant differences among fresh dough samples (*p* < 0.05). Different uppercase letters show statistically significant differences among frozen dough samples (*p* < 0.05).

**Figure 4 foods-12-03845-f004:**
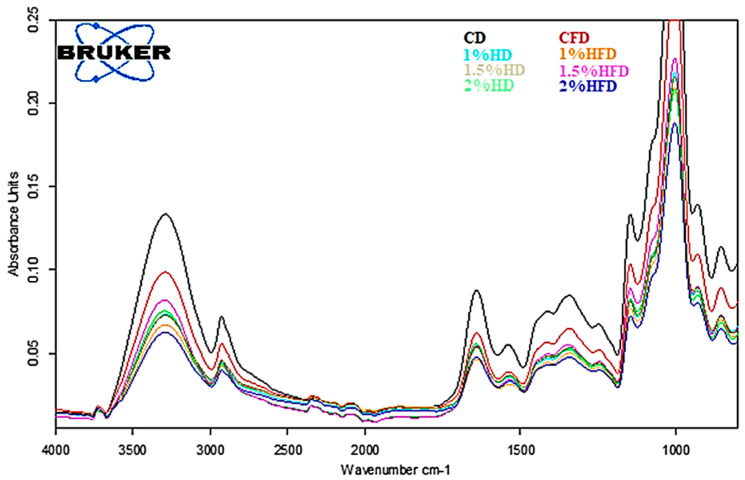
FTIR spectra of the samples containing casein savinase hydrolysate (CSH) under frozen storage. CD: control dough, CFD: control-frozen dough, HD: dough containing CSH, HFD: frozen dough containing CSH.

**Figure 5 foods-12-03845-f005:**
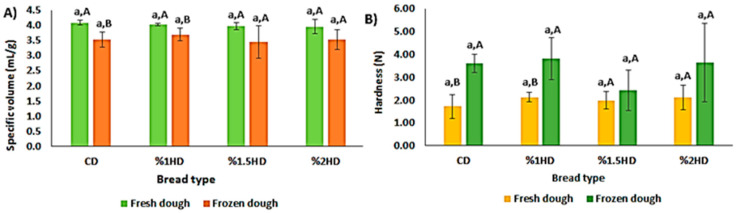
Effect of casein savinase hydrolysate (CSH) addition on specific volume (**A**) and hardness (**B**) of breads. CD: control dough, HD: dough containing CSH. Different lowercase/uppercase letters show statistically significant differences between fresh dough and frozen dough, respectively.

**Table 1 foods-12-03845-t001:** Effects of casein savinase hydrolysate on the secondary structure of gluten protein during frozen storage.

Storage Time	Samples	β-Sheets	β-Turns	α-Helices	Random Coils
	CD	12.42 ± 1.92 ^aA^	28.72 ± 1.82 ^aB^	49.46 ± 3.13 ^aA^	9.71 ± 1.34 ^bA^
Day 0	1% HD	7.75 ± 1.05 ^bA^	23.12 ± 1.51 ^bB^	33.08 ± 2.87 ^bA^	36.05 ± 2.80 ^aA^
	1.5% HD	15.04 ± 1.12 ^aB^	17.38 ± 1.17 ^cA^	35.10 ± 2.63 ^bA^	32.48 ± 3.35 ^aA^
	2% HD	14.14 ± 2.11 ^aA^	19.75 ± 1.43 ^bcA^	32.61 ± 2.64 ^bA^	34.00 ± 2.83 ^aA^
	CFD	16.30 ± 1.29 ^bA^	41.22 ± 1.94 ^aA^	35.53 ± 1.47 ^aB^	6.45 ± 1.85 ^bA^
Day 28	1% HFD	10.24 ± 1.61 ^cA^	33.74 ± 1.82 ^bA^	22.14 ± 2.81 ^bA^	33.88 ± 1.58 ^aA^
	1.5% HFD	23.06 ± 1.44 ^aA^	21.39 ± 2.19 ^cA^	28.05 ± 2.72 ^bA^	27.5 ± 3.54 ^aA^
	2% HFD	19.73 ± 1.53 ^abA^	24.8 ± 2.33 ^cA^	24.24 ± 1.30 ^bA^	31.23 ± 4.40 ^aA^

Data are expressed as mean ± standard deviations of duplicate determinations. Means with different lowercase letters in the same column indicate significant differences (*p* < 0.05) for the samples at different casein savinase hydrolysate levels (CSH). Different uppercase letters show statistically significant effects of frozen storage among the same dough types (*p* < 0.05). CD: control dough; CFD: control-frozen dough, HD: dough containing CSH, HFD: frozen dough containing CSH. The addition levels of CSH are 1%, 1.5% and 2%.

**Table 2 foods-12-03845-t002:** Color properties of breads.

Storage Time	Samples	Crust				Crumb			
		L*	a*	b*	ΔE	L*	a*	b*	ΔE
	CD	58.58 ± 3.59 ^aA^	11.83 ± 0.84 ^aA^	29.34 ± 1.12 ^aA^	-	70.44 ± 2.01 ^aA^	−0.01 ± 0.27 ^aA^	14.81 ± 0.67 ^cB^	-
	1% HD	55.78 ± 2.71 ^abA^	12.87 ± 0.70 ^aA^	31.58 ± 1.53 ^aA^	5.60 ± 3.35 ^aA^	71.45 ± 1.10 ^aA^	0.06 ± 0.05 ^aA^	16.46 ± 0.26 ^abA^	2.99 ± 1.15 ^aA^
Day 0	1.5% HD	52.37 ± 2.94 ^bcA^	13.50 ± 1.23 ^aA^	31.99 ± 0.95 ^aA^	7.09 ± 2.14 ^aA^	72.83 ± 2.16 ^aA^	−0.04 ± 0.13 ^aA^	16.27 ± 0.48 ^bA^	4.07 ± 2.21 ^aA^
	2% HD	49.93 ± 3.67 ^cA^	13.62 ± 1.24 ^aA^	28.40 ± 1.67 ^aA^	9.31 ± 6.17 ^aA^	72.27 ± 0.78 ^aA^	0.05 ± 0.09 ^aA^	17.13 ± 0.51 ^aA^	3.45 ± 1.44 ^aA^
	CFD	58.32 ± 0.77 ^aA^	11.13 ± 0.61 ^aA^	24.73 ± 1.92 ^aB^	6.09 ± 1.91 ^b^	63.76 ± 2.50 ^bB^	0.02 ± 0.14 ^aA^	16.25 ± 0.83 ^aA^	6.92 ± 1.53 ^aA^
Day 28	1% HFD	57.06 ± 2.55 ^abA^	12.31 ± 1.15 ^aA^	24.47 ± 1.81 ^aB^	5.70 ± 2.11 ^bA^	67.90 ± 1.69 ^aB^	0.12 ± 0.23 ^aA^	16.79 ± 1.07 ^aA^	3.46 ± 1.29 ^bA^
	1.5% HFD	55.30 ± 0.71 ^bA^	11.90 ± 2.08 ^aA^	23.01 ± 1.15 ^aB^	8.13 ± 1.84 ^abA^	67.48 ± 3.43 ^abB^	0.22 ± 0.26 ^aA^	16.76 ± 0.29 ^aA^	3.96 ± 2.01 ^bA^
	2% HFD	52.67 ± 1.39 ^cA^	11.83 ± 0.81 ^aA^	22.83 ± 4.07 ^aB^	9.84 ± 2.72 ^aA^	67.90 ± 0.90 ^aB^	0.30 ± 0.03 ^aA^	16.80 ± 0.52 ^aA^	3.66 ± 1.30 ^bA^

Data are expressed as mean ± standard deviations of triplicate determinations. Means with different lowercase letters in the same column indicate significant differences (*p* < 0.05) for the samples at different casein savinase hydrolysate levels (CSH). Different uppercase letters show statistically significant effects of frozen storage among the same bread types (*p* < 0.05). CD: control dough; CFD: control-frozen dough, HD: dough containing CSH, HFD: frozen dough containing CSH. The addition levels of CSH are 1%, 1.5% and 2%.

## Data Availability

The data that support the findings of this study will be made available upon request.

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
