# Peer review of "Effects of Casein Hydrolysate Prepared with Savinase on the Quality of Bread Made by Frozen Dough"

_foods, 2023, doi:10.3390/foods12203845_

Round 1

Reviewer 1 Report

The article presents the results of some research with an impact in the bakery industry, namely the improvement of the quality of bread obtained from frozen dough.

The researches carried out highlighted the beneficial effects of using casein hydrolysate prepared with savinase in frozen dough, the authors showing that this could be an alternative to reduce the deterioration of the frozen bread quality.

The scientific quality of the manuscript it rises to the scientific level of the Foods Journal. The technical quality of the manuscript is good in terms of how it was written and how the experimental results are presented. The style of expression reflects the scientific training of the authors being in accordance with the requirements of writing the article.

The Abstract is concise and contains sufficient information to highlight the content of the article and the Introduction section provides a clear statement of the problem studied in the present manuscript.

The Materials and methods section are well presented and appropriate for the purpose of research.

Results follow the guidelines described in the Author Guidelines and they are well presented and discussed.

References are relevant and current and follow the journal’s format. However, the authors are advised to add more bibliographic references to this section because they are not enough.

The Conclusions of the article are relevant and clearly reflect the results of the study.

Author Response

Thanks for your evaluation.

Regards,

Reviewer 2 Report

Papers looks interesting by means of bread processing knowledge development. The aim and planning of experiment seems to be OK however some drawbacks can be found in methodology section. In general the manuscript may be interesting for the specific group of readers. I designate them as needed minor revision by means of short list of suggestion listed below.

1.       Line 75-76. Please add some more scientific information on the process. Writing “the casein could absorb water better” is far not enough and not scientific

2.       Please add a kneading parameters

3.       Because tan δ = G''/G' the 1c figure or 1aand 1b) are useless

4.       FTIR spectra should be added to the manuscript

Author Response

      We would like to thank you for your valuable assessments about our manuscript. We have completed all the revisions and highlighted them in red within the manuscript. We hope that our manuscript will be accepted after this evaluation.

1

Line 75-76. Please add some more scientific information on the process. Writing “the casein could absorb water better” is far not enough and not scientific

We revised that sentence in line 75-76 as ‘’Casein was first dissolved with distilled water and kept at +4°C for 10 h. for complete hydration.’’. This is the first stage of the procedure of hydrolysis.The details of the procedure were given under this section.
2 Please add a kneading parameters We added the kneading parameter in line 92 [All ingredients were kneaded in a dough mixer (Kitchen aid, USA) at speed 4 for 5 min.] and highlighted. 
3 Because tan δ = G''/G' the 1c figure or 1a and 1b) are useless For frozen dough studies, the evaluation of tan δ value is sufficient. You are right. We also would like to show G'' and G' values in order to determine the behaviour of dough under lineer visco-elastic region and see the effect of casein hydrolysate addition on the rheological behavior.
4 FTIR spectra should be added to the manuscript We added FTIR spectra in section 3.4 as figure 4.

Reviewer 3 Report

Dear Authors,

The current manuscript reports the effects of casein hydrolysate prepared with savinase on the quality of breads made by frozen dough.

In general, this is an important and interesting research, logically structured. The research methods used are described in detail, the results are discussed. References are used appropriately. The results are presented clearly and easily readable.

I have, however, a few comments or suggestions.

Have any studies been previously conducted on the resulting casein hydrolyzate? To what degree of hydrolysis was hydrolysis carried out? Why was this particular technology for its production chosen? Can you please provide links to the relevant sources?

In section 2.9, add the number of experiment repetitions.

The ΔE values turned out to be large, maybe it would make sense to carry out more repetitions of color measurements, and not 3.

Author Response

We would like to thank you for your valuable assessments about our manuscript. We hope that our manuscript will be accepted after this evaluation.

1

1.      Have any studies been previously conducted on the resulting casein hydrolyzate?

2.      To what degree of hydrolysis was hydrolysis carried out?

3.      Why was this particular technology for its production chosen?

4.      Can you please provide links to the relevant sources?

1.      There are many studies related to production, and characterization of casein hydrolyzate (CH). But, the usage of CH into frozen dough was firstly studied in our research.

2.      Thanks for this contribution. The degree of hydrolysis was indicated in the section 2.2 in line 82-83.

3.      Enzymatical hydrolysis prosess is performed by using this method according to Adler-Nissen(1986). This method was cited 2805 times. So, this is widely acceptable method for the production of protein and determination of the hydrolysis degree.There are also several studies which use this method in the literatures:

1- Shazly, A. B., He, Z., Abd El-Aziz, M., Zeng, M., Zhang, S., Qin, F., & Chen, J. (2017). Fractionation and identification of novel antioxidant peptides from buffalo and bovine casein hydrolysates. Food Chemistry, 232, 753-762.

2- Rao, P. S., Mayur, A., Harisha, N. B., Bajaj, R., & Mann, B. (2018). Comparison of OPA and pH stat methods for measurement of degree of hydrolysis of alcalase and flavourzyme digested casein. Indian J Dairy Sci71(1), 107-109.

4. We also added these 2 references in our reference list.

2

In section 2.9, add the number of experiment repetitions.

All experiments have different repetitions in our study. So, we added the experiment repetitions into relevant sections (e.g in line 116, 133, 138, etc.) in the manuscript instead of section 2.9.

3

The ΔE values turned out to be large, maybe it would make sense to carry out more repetitions of color measurements, and not 3.

Thanks for your suggestion. It would be really good if we had a chance to repeat the color measurements. We will consider this for our further researches.